# Passive Attenuation of Mechanical Vibrations with a Superelastic SMA Bending Springs: An Experimental Investigation

**DOI:** 10.3390/s22093195

**Published:** 2022-04-21

**Authors:** Richard Senko, Vinícius S. Almeida, Rômulo P. B. dos Reis, Andersson G. Oliveira, Antonio A. Silva, Marcelo C. Rodrigues, Laura H. de Carvalho, Antonio G. B. Lima

**Affiliations:** 1Laboratory of Vibrations and Instrumentation (LVI), Department of Production Engineering, Federal University of Campina Grande (UFCG), Campina Grande 58428-830, Brazil; 2Laboratory of Vibrations and Instrumentation (LVI), Department of Mechanical Engineering, Federal University of Campina Grande (UFCG), Campina Grande 58428-830, Brazil; 13viniciusalmeida@gmail.com (V.S.A.); antonio.almeida@ufcg.edu.br (A.A.S.); laura.hecker@ufcg.edu.br (L.H.d.C.); antonio.gilson@ufcg.edu.br (A.G.B.L.); 3Department of Technology and Engineering, Rural Federal University of Semi-Árido (UFERSA), Mossoró 59625-900, Brazil; romulopierre@ufersa.edu.br; 4Department of Renewable Energy, Federal University of Alagoas (UFAL), Maceió 57309-005, Brazil; andersson.oliveira@ceca.ufal.br; 5Department of Mechanical Engineering, Federal University of Paraíba (UFPB), João Pessoa 58051-900, Brazil; celoce@ct.ufpb.br

**Keywords:** shape memory alloys, passive dampers, mechanical vibrations, rotor system, experimental, superelasticity

## Abstract

This work presents an experimental study related to the mechanical performance of a special design spring fabricated with a superelastic shape memory alloy (SMA-SE). For the experimental testing, the spring was coupled in a rotor machine, aiming to attenuate the mechanical vibration when the system went through a natural frequency without any external power source. It was verified that the reduction in instabilities stemmed from the better distribution of vibration force in the proposed device, as well as the damping capacity of the spring material. These findings showed that the application of the M-Shape device of SMA-SE for three different cases could reduce vibration up to 23 dB when compared to the situations without, and with, 1.5 mm of preload. The M-Shape device was shown to be efficient in reducing the mechanical vibration in a rotor system. This was due to the damping capacity of the SMA-SE material, and because the application did not require any external source of energy to generate phase transformation.

## 1. Introduction

Shape memory alloys (SMAs) are a class of metallic materials that have two specific characteristics: shape memory effect (SME) and superelasticity (SE). SMA-MEs have the ability to fully recover an apparently plastic deformation introduced in a lower-temperature phase (martensite) by heating it to a temperature high enough to promote a reverse transformation of the material (from martensite to austenite).

SMA-SE is related to the ability of the material to undergo large reversible deformations. This is associated with a phase transformation (from austenite to martensite) induced by an applied mechanical loading and, after load withdrawal, the recovery of the original shape without the occurrence of excessive plastic deformation. These effects are associated with a phase transformation induced by temperature or mechanical stress in SMA, called thermoelastic martensitic transformation [1,2,3,4]. Further, each phase of this material has different properties; for example, austenite has high stiffness and low damping, while martensite has low stiffness and high damping.

Figure 1A shows the SMA-ME phenomena, with a modified shape in the martensitic phase due to an external load. Mechanical loading forces the martensite variants to reorient (detwinning) into a single variant, leading to large macroscopic inelastic strain (from twinned martensite to detwinned martensite). When the load is withdrawn, the material remains in a detwinned state and the inelastic strains are not recovered (steps 2–3). For recovery, it is necessary to heat the material so that the transformation from detwinned martensite phase to austenite phase can occur. After recovery, as the material cools, it returns to the twinned martensite phase without residual deformation.

Figure 1B illustrates the full phase transformation, from the austenite phase to the detwinned martensite phase, which is directly produced in SMA–SE with the application of an external load in segments (steps 1–2–3). This process is associated with large inelastic strains, which are recovered upon unloading (steps 3–4–1) due to the reverse transformation. The complete load–unload cycle results in a hysteresis curve, where the area inside these curves represents the dissipated mechanical energy in the cycle [3,4,5,6,7].

The ability of an SMA to return its original state after some deformation, and the difference in the stiffness and damping of each involved phase, is the reason for the extensive studies on the use of this class of material in order to reduce vibration in rotor systems [8,9]. For this application, both effects (SME and SE) have been frequently used. SMA-ME has been used in many types of designs, such as helical springs [9,10,11,12,13], wires [14,15], and special designs [16]. For SMA-SE, the number of applications has also been growing lately—particularly, applications in helical springs [17,18,19], wires [2,20,21,22,23], and special designs [24].

However, applications with SMA-ME are limited because it is thermally active, requiring an external source of energy to provoke material phase change through heating and cooling. In addition, the thermally active material exhibits a low-frequency response (up to 10 Hz), as reported in the literature [9]. Moreover, SMA-ME requires additional equipment. In order to change the temperature of the SMA-ME device, a control system is required to precisely activate the heating process, while a cooling system is required to recover the previous phase of the material. In order to improve the efficiency of this controller, it is usually necessary to add a timer to change the temperatures of the device to achieve the complete phase transformation or use an efficient piece of equipment to execute the process, which increases the cost. Differently, SMA-SE is activated by mechanical stress and is capable of operating in a frequency range that exceeds 10 Hz, without the need for an external power source. Moreover, under test conditions, this type of effect responds faster when submitted to mechanical stress than SMA-ME [4].

Therefore, knowing the characteristics of SMA-SE and focusing on new types of spring designs, following the simulations carried out in our previous work [25] and as a compliment to the existing research, this paper describes an experimental study on the mechanical performance of a spring device with an innovative special design (M-Shape) made of SMA-SE. Our intention is to reduce mechanical vibrations in a rotor system for different applications.

## 2. Materials and Methods

### 2.1. Design of Bending Spring M-Shape Device

The bending spring design (M-Shape) shown in Figure 2A [26] focused on obtaining a larger contact area with the bearing to provide stability for the rotor system acceleration and create a better way to distribute the force applied by the ball bearings and shaft, as illustrated in Figure 2B.

Due to its geometry, this device has a low stiffness. This is an important characteristic because high stiffness levels can reduce the instabilities, changing the natural frequency of the system [27], or, as a result, changing it during the application [9,13]. The vibration attenuation due to stiffness shifting is not within the scope of this paper. 

This M-Shape device was based on the bending spring design presented in our previous work [25,26]. It has fewer curves to facilitate fabrication and to avoid major changes in the phase transformation temperatures of the SMA-SE material, which mainly depend on its composition and history (fabrication process, heat treatments, and others) [4,7,28].

### 2.2. Fabrication of the M-Shape Device

A NiTi plate (ASTM F2063-2000) with a thickness of 0.3 mm (produced by Sunrise Titanium Technology Co. Ltd., Shaanxi, China) with 55.57% of Ni was used to manufacture the M-Shape device with the dimensions shown in Figure 2A. The first step was cutting the NiTi plate, with the dimensions shown in Figure 3, using an electric discharge cutting machine (EDM). 

Then, a heat treatment (500 °C for 60 min with cooling by natural convection), as indicated by the manufacturer of the NiTi plate, was applied for homogenization and stress relief [29]. Following this, a rolling process with reduction steps of 10% was conducted in order for the NiTi strips to reach sizes of 0.25 mm. Between each step, another heat treatment (450 °C for 20 min, with cooling by natural convection to relieve stress) was applied. These experimental procedures were proposed so as not to compromise or cause only minimum damage in the microstructure of the material. The aim was to keep the material’s superelasticity and not significantly change its phase transformation temperature. Moreover, according to the literature [30,31], rolling increases plastic deformation resistance as well as the superelastic effect, characteristics that benefit specific applications. To reach the requested thickness, two steps were needed, as shown in Table 1.

The next step was setting the shape into the M-Shape design. In this procedure, the NiTi strip was placed inside the metallic mold (Figure 4A), which was fixed with bolts; cold formed before heating at 500 °C for 30 min; and cooled by natural convection [32,33,34]. The M-Shape device of SMA-SE, as shown in Figure 4B, was then obtained.

### 2.3. Steady Experiment with the M-Shape Device

Measurements of the electrical resistivity as a function of temperature (ERFT) (Figure 5) were conducted using a Huber^®^ CC-902 refrigeration bath circulator in order to obtain and compare the differences in the phase transformation temperatures of the NiTi strip and M-Shape device. Two samples were used: (a) the NiTi strip with the heat treatment proposed by the manufacturer and (b) the SMA-SE manufactured M-Shape device proposed in this work.

The temperatures were acquired by welding four copper wires in each of the samples (1), and connecting them with a previously programmed DC power supply (3) attached in the electrical interface (4). A K-type thermocouple (5) was attached to the sample (1) and linked in the signal conditioner (6) and then to an acquisition system (7). Both samples were conditioned in the refrigeration bath circulator (2). Conditioning took place by thermal cycling from −60 °C to 100 °C in steps of 3 °C/min, following experimental procedures as reported in the literature [35,36].

### 2.4. SEMD (System for Estimation of Material Damping) Experiment

The damping capacity and the effective stiffness of the M-Shape device were analyzed by dynamic testing in SEMD (System for Estimation of Material Damping) developed in our research group [37]. In this experiment, Figure 6, a displacement is applied, using the shaker (item 9), directly in the SDOF (Single Degree of Freedom) system (item 6), which has 1.485 kg, without the SMA-SE device. Following, the M-shape device (item 4) was placed in the position presented in Figure 6, and the same procedure was applied.

The difference between the damping of the SDOF system with and without the SMA-SE device under test, provides an estimate of the material damping, through the experimental FRF receptance curves, and Nyquist diagram. All details about the procedure of the SEMD are presented in the literature [37].

### 2.5. Dynamic Experiment with the M-Shape Device–MTS

Prior to attaching the M-Shape devices in the rotor system, the device was assembled in the support bearing. Then, a dynamic experiment was performed in a universal testing machine (MTS 810 100 kN), as shown in Figure 7.

The M-Shape device was fixed on the support bearing using the pressure of the bearing with the shaft. Additionally, acrylic plates were applied on each side of the support bearing to avoid axial displacements (Y axis). Three physical situations were tested: (a) without preload, (b) with a 1.0 mm preload (displacement), and (c) with a 1.5 mm preload (displacement). These values were defined based on previous work [25,27]. In order to make it easier to assemble the support bearing with the M-Shape device, steel bushing with a ball bearing was used to apply the selected preloads in the rotor system, as shown in Figure 8.

Experiments were conducted with 100 cycles, with displacements of 2 mm from peak to peak at a 1 Hz frequency (Z-axis). Results were reported as curves of force vs. displacement in order to compare the hysteretic loop for each case.

### 2.6. Rotor System with the M-Shape Device of SMA-SE Applied

The M-Shape device was assembled in the support bearing. This system was placed in the rotary machine (Figure 9) to reduce mechanical vibrations due to passage through the natural frequency in passive form.

This rotor system had the following parts: a motor *WEG W22* plus with 3 kW power, a variable-frequency drive inverter *WEG^®^ µline^®^* with a maximum rotation of 3600 RPM (60 Hz), a disk, a shaft, and a support bearing. The parameters of this system are described in Table 2. In this table, ∅_In_ represents the inner diameter and ∅_Out_ is the outer diameter.

In order to obtain the displacements, three proximity sensors (SKF CMSS 665) were inserted in the following locations: two in the highest vibration point in the shaft, near the disk, and another used as a tachometer, as shown in Figure 10. A K-type thermocouple was also attached to the M-Shape device to track temperature changes during the rotor system run up.

Experiments were performed under three experimental conditions—i.e., without and with preloads of 1.0 mm and 1.5 mm in the support bearing with the M-Shape device. An acceleration from 0 to 50 Hz over 500 s (0.1 Hz/s) was applied to the rotor system. This acceleration was chosen because the system should not pass through its natural frequency quickly, which can be a solution to avoid high amplitudes. The average results were obtained in five experiments for each preload.

The experiments were also conducted with temperatures stabilized in a range, after the expected self-heating suffered by the device according to each applied stress level. This difference between the starting temperatures corroborates the values reported in the literature [3,7,37], which shows that the temperature in the material increases with a higher applied stress. However, the analysis of the self-heating of SMA-SE devices is not part of the scope of this study. The results were individually shown and compared to each other, aiming to determine the system’s differences in attenuation.

## 3. Results and Discussions

### 3.1. Bending Spring Characterization

After the fabrication of M-Shape bending springs following the steps shown in the methodology, ERFT measurements were performed in order to identify the temperatures of phase transformation of the SMA-SE material, as shown in Figure 11 and Table 3. In Figure 11, *M*, *A*, and *R* represent martensite, austenite, and R-phase, respectively, and the subscripts *s* and *f* correspond to the initial and final transformation temperatures (TT). 

Upon analysis, Figure 11 and Table 3 reveal that after the fabrication (rolling and shape setting), the material’s phase transformation temperatures were affected in a range of approximately 10 °C according to the specialized literature [4,7,28]. Consequently, the NiTi bending spring does not have the full damping capacity of the superelastic effect in environment temperatures, which should be an issue for passive applications. However, according to Patoor et al. [7], the damping capacity of SMA-SE depends on the excitation frequency, amplitude, temperature, and the difference between the operating and transformation temperatures. In general, there are three distinguished situations of damping with SMA-SE: at temperatures higher than the *M_s_* it has a small damping capacity, at temperatures below the *M_f_* the damping capacity is increased, and at temperatures above the *A_s_* the maximum damping capacity is achieved [7].

The stress generated due to the vibrations of the rotor system leads to device self-heating, reaching temperatures above the *A_f_*, which corroborates the findings of Reis et al. [37]. Self-heating happens in the forward transformation (from austenite to martensite), which is exothermic, different from the reverse transformation (from martensite to austenite), which is an endothermic phase transformation followed by the absorption of thermal energy [7].

### 3.2. SEMD Experiment with the M-Shape Device

Dynamic tests with the M-Shape device in SEMD were performed to analyze the hysteresis damping (h_s_), loss factor (η), and effective stiffness (K_s_). The experimental procedure adopted was that described by Reis et al. [37], and the results obtained are shown in Figure 12 and Table 4.

Here, the h_sma_ is the hysteresis damping of the M-Shape device, K_sma_ is the stiffness of the M-Shape device, and η_sma_ is the loss factor of the M-Shape device.

Upon analysis of Figure 12, it can be verified that there was a drop of –14 dB in the amplitude values of the FRF in comparison to the same situation without the M-Shape device. Furthermore, the hysteresis damping was obtained by analyzing the Nyquist plot. An approximate 80% increase in damping was achieved with the use of this device, as observed by the reduction in the Nyquist’s circle of the experiments, and in Table 4, with and without the M-Shape device. This is taken as an indication that the proposed bending-spring can be used as a damper and effectively reduces vibrations.

### 3.3. Dynamic Test with M-Shape Device Applied in the Support Bearing

After the identification of the temperatures of phase transformation and defining the necessary preloads, dynamic tests were performed with the support bearing with the M-Shape device. The hysteresis loop of each case was determined, without and with a preload of 1.0 mm and 1.5 mm (see Figure 8).

Figure 13 shows the hysteresis loop that was obtained for the M-Shape device tested under the three previously defined conditions according to Patoor et al. [7]. Additionally, calculating the areas of each one of the curves, it can be noticed that the difference between the cases with a preload of 1.0 mm and without is 45%, and that between the preloads of 1.5 mm and 1.0 mm is 18%. These results are in agreement with the literature [25,27], where it was shown that, at a 1.0 mm displacement, martensitic transformation was initiated in the bending springs. Furthermore, it was shown that the hysteretic loop increases with further addition in amplitude. In summary, as the preload applied in the M-shape device increases the damping, it will be able to reduce undesirable vibrations.

### 3.4. The M-Shape Device Applied in the Rotor System Analysis

The experimental results obtained with the superelastic SMA bending springs applied in a rotor system were analyzed under three situations: without preload, and with two different preloads (1.0 mm and 1.5 mm). Consequently, the curves of amplitude vs. time, temperature analyses, FRF, and spectrograms were obtained. All were presented on the same scale and in the following sections.

#### 3.4.1. SMA-SE M-Shape Device without Pre-Load

For this setup, experiments were performed without the preload in the ball bearing (item A in Figure 8). Figure 14 shows the different transient results obtained in the experiments without preload.

Figure 14A,B shows that the temperature of the M-Shape device does not reach the *A_f_* temperature. Therefore, the SMA material could not reach the full damping capacity of the superelastic behavior. This happened because the vibration transmitted from the system to the bending spring was insufficient for the SMA material to start the martensitic transformation. Despite this condition, Figure 14C,D show high amplitudes and levels of energy in the natural frequency region, which is at about 25 Hz.

These results show that, in the area of the highest amplitude, the temperature of the M-Shape device started to increase, indicating the need for a higher vibration amplitude, which is not desirable, or for some load to be applied for the martensitic transformation to occur.

#### 3.4.2. SMA-SE M-Shape Device with Pre-Load of 1.0 mm

For the experiment in the SMA–SE M-Shape device preloaded, a 1.0 mm preload was applied with the ball bearing, as shown in item B in Figure 8. In Figure 15 is illustrated the different transient results obtained in the experiments with a preload of 1.0 mm.

Figure 15A,B indicate that, because the applied preload added to the amplitude of the vibration, the temperature of the M-Shape device reached the final austenitic temperature of transformation (*A_f_*). This behavior proved that the device is capable of displaying an increase in damping capacity of the superelastic effect in this setup, as shown previously in Figure 13. In addition, the increase in temperature led to the same behavior observed in the experiment without the preload, where temperature started to rise in the natural frequency region (~25 Hz) of the rotor system. As a consequence of the preload, the amplitude was reduced by approximately 50%, as seen in Figure 15C. Compared with Figure 14C, increments in temperature occurred which were in concordance with those reported in the literature [4,7,28,37].

Analyzing the results presented in Figure 15D and comparing them with those in Figure 14D, it can be verified that the density of energy around the natural frequency (between 200 s and 300 s) decreased, as expected, due to the increase in the damping capacity of the material superelastic behavior.

#### 3.4.3. SMA–SE M-Shape Device with Pre-Load of 1.5 mm

For the experiment with the preloaded SMA-SE M-Shape device, a 1.5 mm preload was applied with the ball bearing, as shown in Item C in Figure 8. In Figure 16, the different transient results obtained in the experiments with a preload of 1.5 mm are illustrated.

Upon the analysis of Figure 16B, it is evident that, as in the previous experiment, with the application of a 1.5 mm preload, the M-Shape device was able to reach the *A_f_* temperature and display superelastic behavior and, as shown in Figure 13, was expected to have more hysteresis damping. This led to a 65% decrease in the amplitude of the rotor system, as illustrated in Figure 16C, as compared with the results obtained in the previous experiment with a 1.0 mm preload.

Figure 16D confirms this decrease as occurring in the area of the natural frequency. The amount of energy shows few densities, which differs from the cases without and with a 1.0 mm preload. Figure 17 depicts the FRF of the system in all three setups. Upon the analysis of this figure, it is evident that, for the test with the highest preload (1.5 mm) vs. that without preload, the amplitude of vibration was reduced by about 23 dB in the resonance zone (~25 Hz).

This result plays an important role in the vibration field, because it shows that the application of preloads approximates the M-shape device of the full martensite transformation, increasing the damping capacity and leading to reduced vibration in the rotor system. In summary, the system does not need to vibrate at a high level to initiate the martensitic transformation in the bending spring, and then to begin the reduction in instability. In systems with low-level vibrations, the M-Shape device with the right preload will increase hysteretic damping in the rotor system and, consequently, reduce instabilities.

## 4. Conclusions

This paper presents a special design for a bending spring, which is a 0.72 g M-Shape device manufactured with SMA-SE alloy and applied in a rotor system. The strategy is to reduce the mechanical vibrations while the mechanical system passes through the natural frequency. Before the M-Shape device application in the rotor system, static and dynamic experiments were performed in order to identify the temperatures of phase transformation and demonstrate the capability of the damping capacity of this material. From the analysis of the result, it can be concluded that:(a)In the SEMD experiment, proposed by [37], the addition of the M-Shape device of SMA-SE in the SDOF system was able to attenuate the amplitude of the movement by 14 dB. With only 0.048% of the mass of the system, the proposed device added 80% of hysteresis damping, increasing from 289.25 N/m to 1485.7 N/m, demonstrating an excellent damping capability.(b)In the dynamic experiments, an increase in the preload applied in the device of SMA-SE provoked the enhancement of the damping capacity, corroborating the results seen in the literature.(c)At a maximum fixed preload (1.5 mm amplitude), the M-Shape device was able to attenuate the vibrations by −23 dB as applied in the rotor system, when compared with the setup without a preload.(d)The application of superelastic SMA materials in passive control can be effective to achieve better responses of vibration attenuation without reducing the performance of the rotor system, confirming the innovative aspect of the proposed SMA-SE M-Shape device.

## Figures and Tables

**Figure 1 sensors-22-03195-f001:**
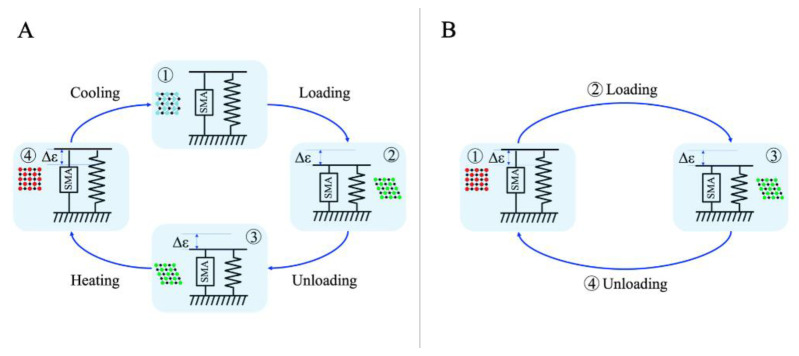
Deformation cycle of shape memory alloys. (**A**) SMA-ME; (**B**) SMA-SE.

**Figure 2 sensors-22-03195-f002:**
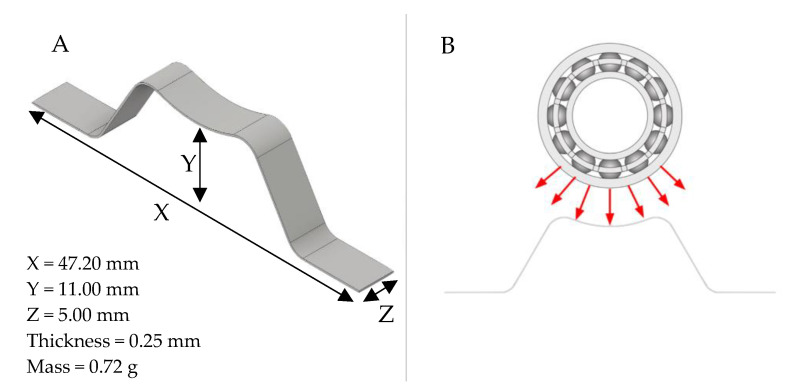
(**A**) M-Shape device of SMA-SE; (**B**) distribution of the forces on the M-Shape device.

**Figure 3 sensors-22-03195-f003:**
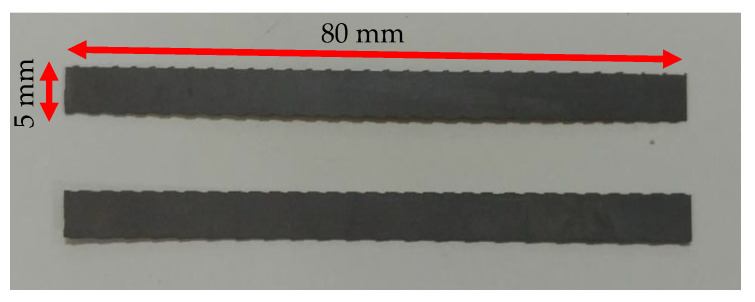
Strips of NiTi SMA-SE.

**Figure 4 sensors-22-03195-f004:**
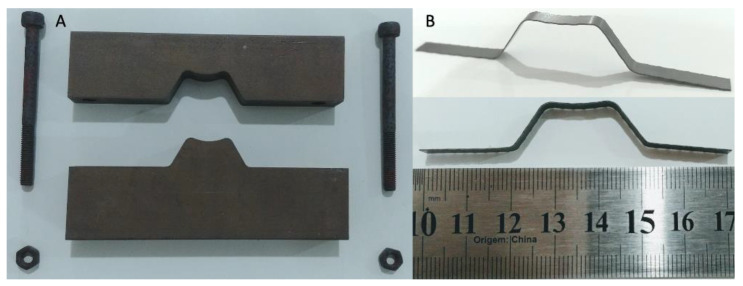
(**A**) Metallic mold used to apply the shape setting process; (**B**) M-Shape device of SMA-SE.

**Figure 5 sensors-22-03195-f005:**
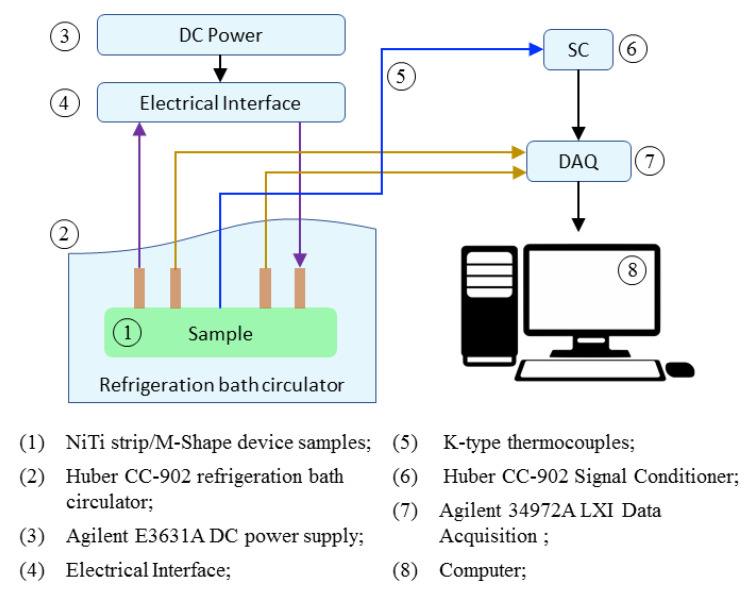
Schematic of the ERFT experiment.

**Figure 6 sensors-22-03195-f006:**
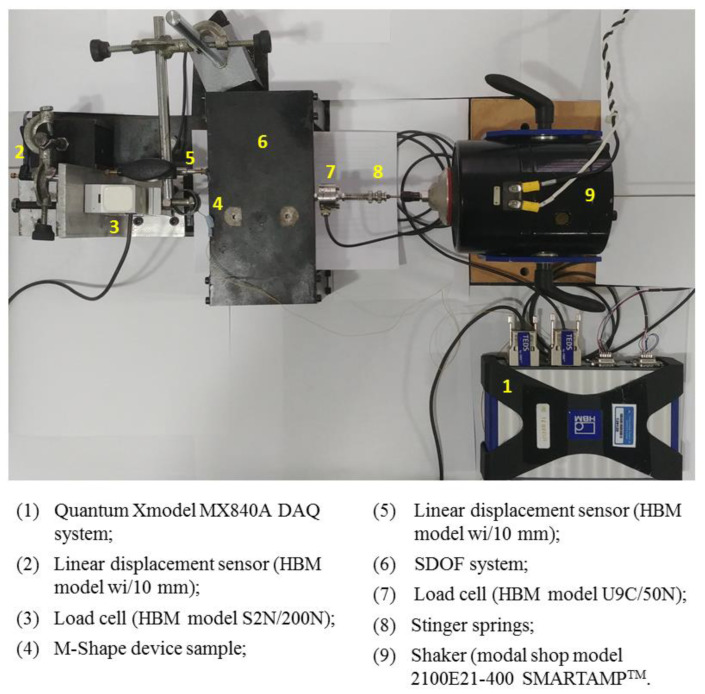
Photograph of the equipment used in the SEMD experiment.

**Figure 7 sensors-22-03195-f007:**
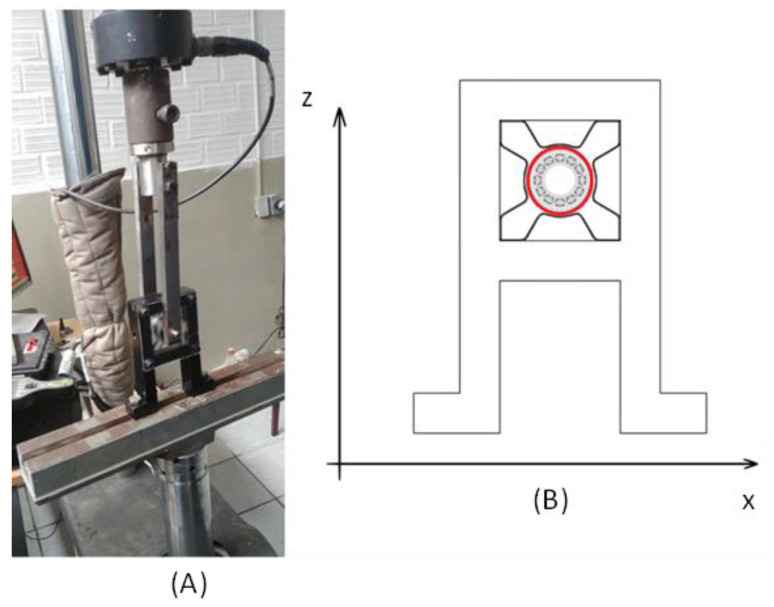
(**A**) Photograph and (**B**) sketch of the dynamic experiment in MTS 810.

**Figure 8 sensors-22-03195-f008:**
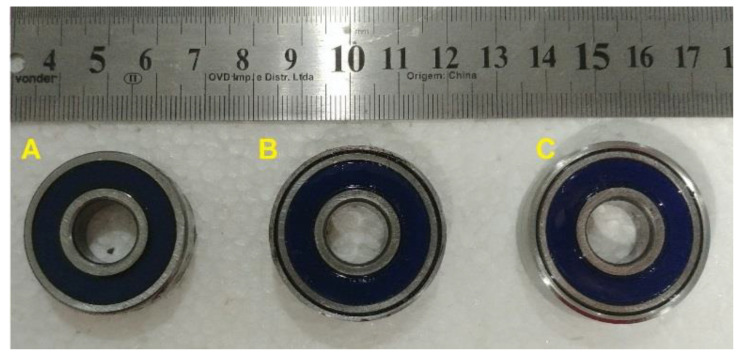
Steel bushing with ball bearings: (**A**) without preload; (**B**) preload of 1.0 mm; (**C**) preload of 1.5 mm.

**Figure 9 sensors-22-03195-f009:**
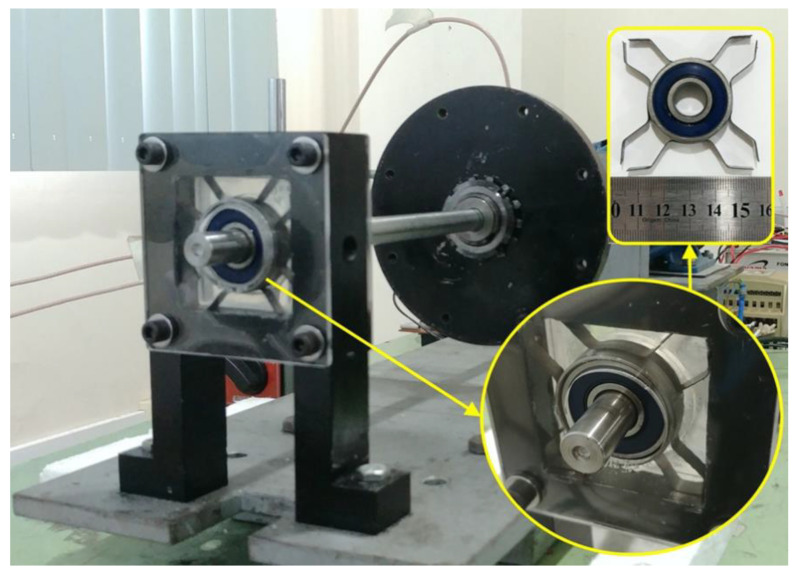
Experimental bench with the M-Shape device.

**Figure 10 sensors-22-03195-f010:**
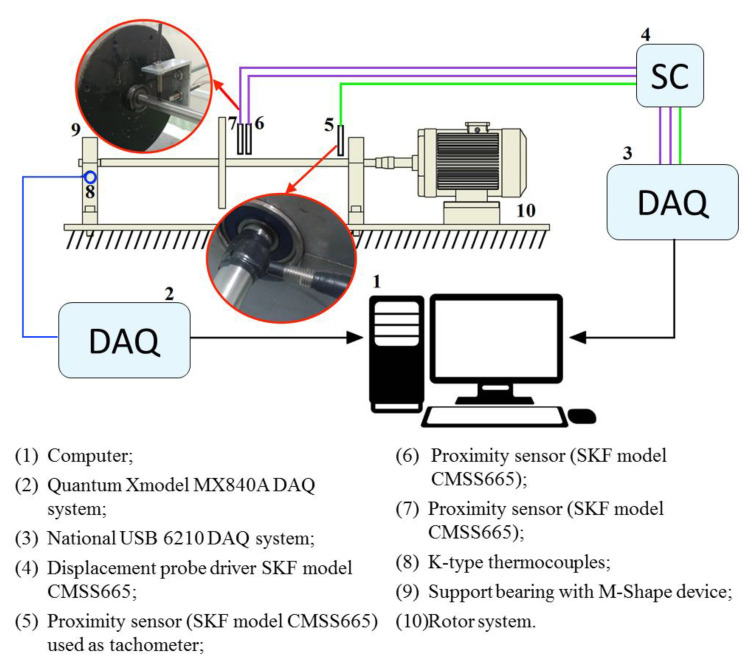
Sketch of the rotor system coupled with a sensor.

**Figure 11 sensors-22-03195-f011:**
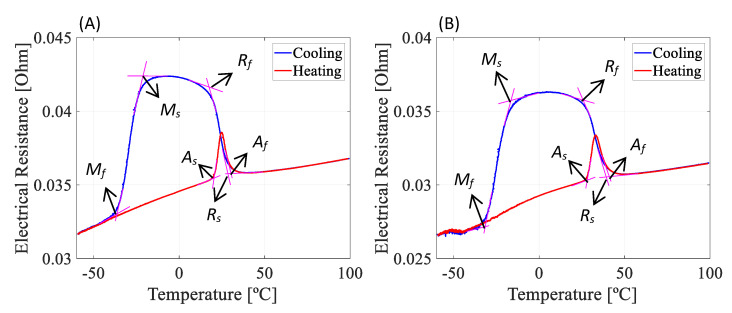
Electrical resistance as a function of the temperature. (**A**) NiTi strip and (**B**) M-Shape device of NiTi.

**Figure 12 sensors-22-03195-f012:**
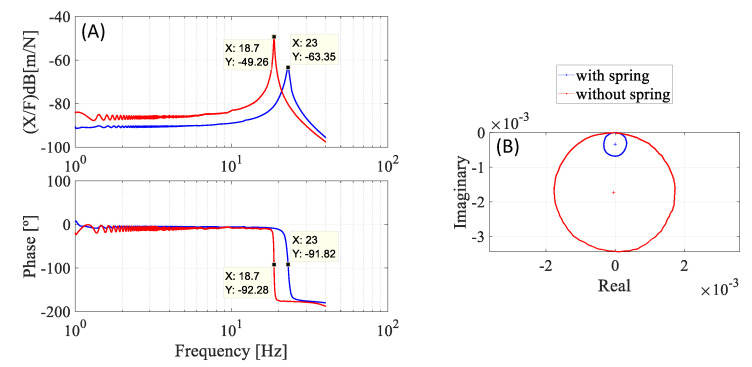
SEMD results of SMA-SE bending spring. (**A**) Bode plots; (**B**) Nyquist plots.

**Figure 13 sensors-22-03195-f013:**
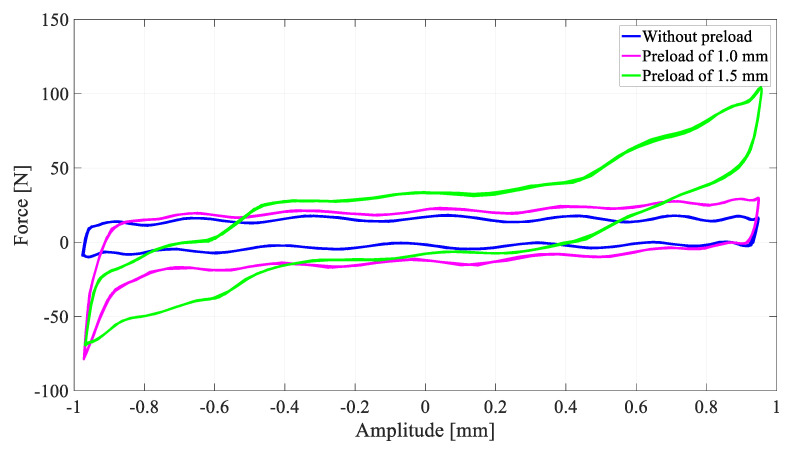
Force versus amplitude for dynamic test with the M-Shape device applied in the support bearing.

**Figure 14 sensors-22-03195-f014:**
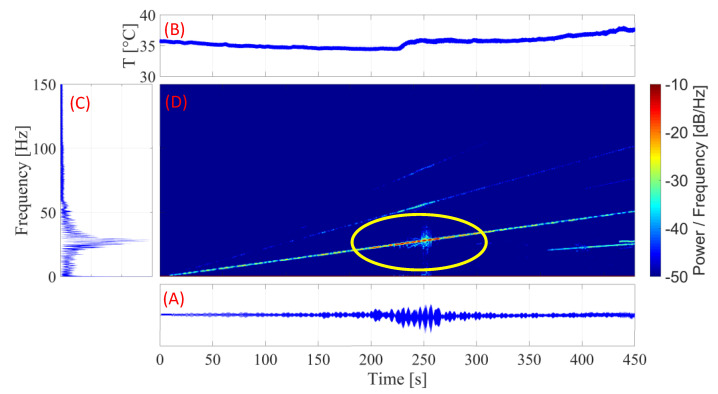
Curve for the SMA-SE M-Shape device without pre-load: (**A**) amplitude (peak to peak) × time; (**B**) temperature × time; (**C**) FFT; (**D**) spectrogram.

**Figure 15 sensors-22-03195-f015:**
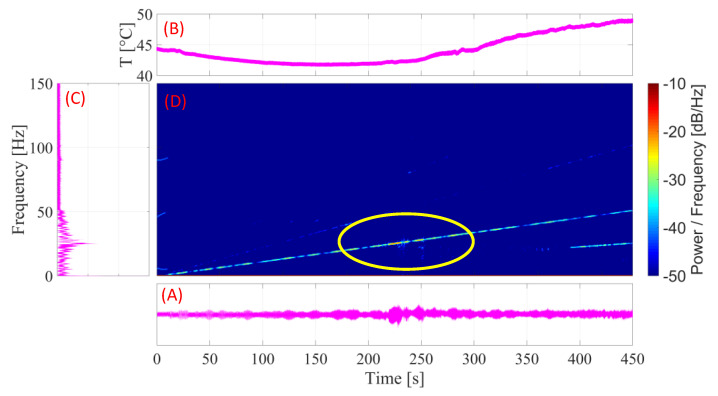
Curves for the SMA-SE M-Shape device with a pre-load of 1.0 mm: (**A**) amplitude × time; (**B**) temperature × time; (**C**) FFT; (**D**) spectrogram.

**Figure 16 sensors-22-03195-f016:**
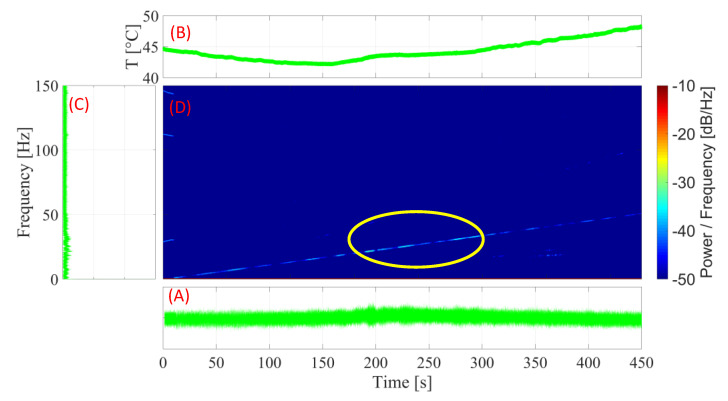
Curve for the SMA-SE M-Shape device with pre-load of 1.5 mm: (**A**) amplitude × time; (**B**) temperature × time; (**C**) FFT; (**D**) spectrogram.

**Figure 17 sensors-22-03195-f017:**
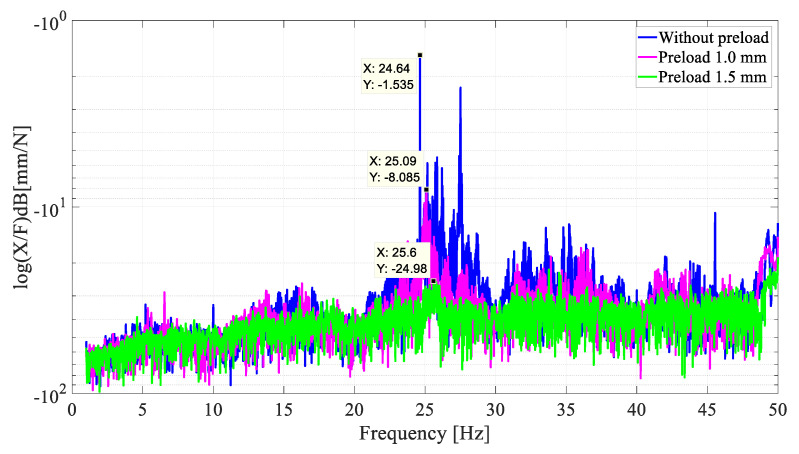
FRF of the system with and without preload.

**Table 1 sensors-22-03195-t001:** Steps of the rolling process.

Step	T_s_ (mm)	10% of Thickness (mm)	T_f_ (mm)
1	0.30	0.03	0.27
2	0.27	0.02	0.25

**Table 2 sensors-22-03195-t002:** Experimental bench parameters.

	Mass (kg)	Ø_In_ (mm)	Ø_Out_ (mm)	Length (mm)	Width (mm)
Shaft	0.4816	–	12.50	414.00	–
Disc	1.4335	26.00	150.00	–	11.00
Support Bearing	2.096	37.00	–	79.00	25.00

**Table 3 sensors-22-03195-t003:** Temperatures of phase transformation of the SMA-SE.

Phase Temperature	*M_s_* (°C)	*M_f_* (°C)	*R_f_* (°C)	*R_s_* (°C)	*A_s_* (°C)	*A_f_* (°C)
NiTi Strip–(A)	−35.19	−23.14	19.48	28.70	20.63	30.39
M-Shape of NiTi–(B)	−31.32	−17.31	26.55	39.22	28.09	40.53

**Table 4 sensors-22-03195-t004:** Hysteresis damping, stiffness, and loss factor obtained from SEMD.

	A SDOF without M-Shape Device	B SDOF with M-Shape Device	Unit
h_s_	289.25	1485.7	N/m
h_sma_ ^a^	-	1196.45	N/m
K_s_	18,381	33,014	N/m
K_sma_ ^b^	-	14,633	N/m
η_s_	0.0156	0.1313	-
η_sma_ ^c^		0.1157	-

^a^—h_sma_ = (B − A) with row h_s_. ^b^—K_sma_ = (B − A) with row K_s_. ^c^—η_sma_ = (B − A) with row η_s_.

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
