# Peer review of "Passive Attenuation of Mechanical Vibrations with a Superelastic SMA Bending Springs: An Experimental Investigation"

_sensors, 2022, doi:10.3390/s22093195_

Round 1

Reviewer 1 Report

Page 1, line 32  Should the abbreviation of Shape Memory Effect be SME?

Page 3, lines 109 to 114   How does this process ensure the SE of the material and ensure that the phase transformation temperature does not have significantly change?

Author Response

Thanks for your review. We used this abbreviation, ME for Shape Memory Effect, in the previous paper [25], we thought just to keep the pattern. But we changed to SME as you suggested (p.1 line 32 and p.2 line 63).

Reviewer 2 Report

In this manuscript, the attenuation of mechanical vibrations using springs made of an alloy showing superelasticity.  The results are reasonable, and I have found no serious mistakes in the manuscript.  I think this manuscript can be accepted after native-English check.

Author Response

Thanks for your review, and as you suggested we made another revision in the text.

Reviewer 3 Report

This study presents a special design of bending spring M-Shape device manufactured with SMA-SE alloy and applied in a rotor system. The temperatures of phase transformation and demonstrated the capability of the damping capacity of this material, then the application of the M-Shape device in the rotor system, were performed static and dynamic experiments.The experiments are systematic and the results are instructive for the application of shape memory alloy in damping structure design and engineering. The manuscript is well written.For these reasons I think that this work is suitable to be published in sensors after minor revisions.

Comments:

  • Page 9 figure The figure description for each subgraph (A-D) should be marked in the legend. Figure 14-16 all have the same problem.
  • Figure 14-16. Is the test ambient temperature room temperature? Which temperature point is the temperature in the figure? Why did the starting temperature be different in the three trials? How to control the experimental temperature?
  • The three conclusions need to be reorganized and do not fully reflect the innovation points of the content of the article.

Author Response

Thanks for your review. The answers to your comments are below:

  • Page 9 figure The figure description for each subgraph (A-D) should be marked in the legend. Figure 14-16 all have the same problem.

Ans - We changed the legend of each figure (14-16) to: (A) Amplitude (peak to peak) x Time; (B) Temperature x Time; (C) FFT; (D)Spectrogram.

  • Figure 14-16. Is the test ambient temperature room temperature? 

Ans - The experiments were carried out at room temperature, between 23ºC and 25ºC.

  • Which temperature point is the temperature in the figure? 

Ans - The temperature shown in Figures 14-16 is from the SMA-SE spring applied to the rotor system, where they were obtained through a K-type thermocouple, as shown in Section 2.6 and Figure 10 (p.7). This information has the objective of showing that the proposed SMA-SE device is (case with preload) or not (case without preload) within the phase transformation temperature range.

  • Why did the starting temperature be different in the three trials? 

Ans - Starting temperatures differ between each experiment, due to self-heating and stabilization. In the case of springs without preload, it was noticed that there was no considerable self-heating, stabilizing out of the range of austenitic transformation temperature. This is due to the low deformation that the set of springs applied to the system suffers, only the action of the amplitude of vibration of the system.

For springs with applied preloads of 1mm and 1.5mm, as soon as the system passes through the natural frequency, they suffer a greater deformation, due to the action of the amplitude of vibration + the preload, therefore with higher applied stress the temperature of the device increases (through an exothermic reaction) to within the phase transformation temperature range.

In the attached figures, it can be noted that Run 1 of the figures with preloads increases the temperature in the time range of 230 to 300 seconds (area of ​​the rotor system's natural frequency). However, as the analysis of the self-heating of the SMA-SE devices was not part of the scope of the study, it was decided to carry out the experiments at the temperatures at which the springs stabilized.

This difference in applied stresses was already expected, as seen in figure 13 (p. 9) (a dynamic experiment in MTS), and during the design phase, the need to apply preloads to obtain higher deformations was already being predicted in simulations, therefore greater forces are applied, as can be seen in [25].

Also, this behavior corroborates with the literature [3], which shows that with increasing applied stress, an increase in temperature occurs, consequently in higher temperature ranges the austenite is more stable and higher stress is required for stress-induced martensite (detwinned martensite). This is in accord with the Clausius-Clapeyron relationship (eq 1) since the enthalpy of transformation is exothermic (which occurs due to self-heating) for stress-induced martensite.

dσ/dT=-ΔS/ε=ΔH*/εT                           (1)

Where σ – Uniaxial stress; ε – Transformations strains; ΔS – Entropy of transformations per unit of volume; ΔH* - Enthalpy of the transformation per unit of volume.

  • How to control the experimental temperature?

In this case, to control the temperature of the experiments, it would be necessary to apply thermal blowers, application of the Joule effect, and coolers to stabilize the temperatures in the sought ranges. As seen in dos Reis 2020 [37] in a thermomechanical experiment with a mini helical spring of SMA-SE or Oliveira 2020 [13] that applied this temperature control in helical SMA-ME springs.

  • The three conclusions need to be reorganized and do not fully reflect the innovation points of the content of the article.

Ans - Conclusion

This paper presents a special design of bending spring, a 0.72g M-Shape device manufactured with SMA-SE alloy and applied in a rotor system. The strategy is to reduce the mechanical vibrations during the mechanical system passing through the natural frequency. Before the M-Shape device application in the rotor system, were performed static and dynamic experiments, to identify the temperatures of phase transformation and demonstrated the capability of the damping capacity of this material. From the analysis of the result, it can be concluded that:

a) In SEMD experiment, proposed by [37], demonstrated the addition of the M-Shape device of SMA-SE in the SDOF system was able to attenuate the amplitude of the movement by 14dB. With only 0.048% of the mass of the system, the proposed device added 80% of hysteresis damping, increasing from 289.25 N/m to 1485.7 N/m, demonstrating an excellent damping capability.

b) In the dynamic experiments, an increase in the preload applied in the device of SMA-SE provoked an enhancement of the damping capacity, corroborating with the literature.

c) At a maximum fixed preload (1.5 mm amplitude), the M-Shape device was capable to attenuate the vibrations by -23 dB as applied in the rotor system, when compared with the setup without preload.

d) The application of superelastic SMA materials in a passive control can be effective to achieve better responses of vibration attenuation without reducing the performance of the rotor system, confirming the innovative aspect of the proposed SMA-SE M-Shape device.
